# n77 Radio Frequency Power Amplifier Module for 5G New-Radio High-Power User Equipment Mobile Handset Applications

Ji-Seon Paek

Department of Electronics Engineering, Pusan National University, Busandaehak-ro 63beon-gil, Gumjeong-gu, Busan 46241, Republic of Korea; js.paek@pusan.ac.kr

**Abstract:** This paper presents a highly efficient 5G New-Radio (NR) RF power amplifier module (PAM). The n77 PAM consists of a high-voltage differential-topology 2 μm GaAs HBT power amplifier, a CMOS controller, a silicon-on-insulator (SOI) switch, an integrated passive device (IPD) bandpass filter, a low-noise amplifier (LNA), and a bi-directional coupler. This PAM generates a saturation output power of 32.7 dBm including the loss of the SOI switch and output filter. The designed n77 PAM is tested with a commercial envelope tracker IC (ET-IC). The designed PAM with an ET-IC achieves an ACLR of −37 dBc at a 27 dBm output power with a DFT-s-OFDM QPSK 100 MHz NR signal and saves a dc power consumption of 950 mW compared to the APT mode. For the CP-OFDM 256QAM with the most stringent EVM requirements, it achieves an EVM of 1.22% at 23 dBm and saves 640 mW compared to the APT mode.

**Keywords:** cellular mobile handset; envelope tracking (ET); high-power user equipment (HPUE); new radio (NR); power amplifier module (PAM)

## 1. Introduction

Generally, the RF power amplifier (RF-PA) in new NR bands suffers from poor efficiency due to the high-frequency operation, higher peak-to-average ratio (PAPR) of the CP-OFDM, and loss of the RF front-end module to support complex operating band combinations (EN-DC for non-standalone mode, 2CA/3CA). The poor PA efficiency can be improved with a 100 MHz envelope tracking (ET) [1–8] operation and high-voltage PA design above 5 V. Although ET [1–23] is a key technology that improves the battery usage time of mobile handsets by modulating the RF-PA supply voltage according to the signal envelope, there are still several technical challenges to applying ET technology to a 5G mobile handset while supporting the channel BW of 100 MHz and the PC2 power of 26 dBm. To address these challenges, this work designed a highly efficient 5G NR n77 RF power amplifier module. To achieve the best performance, we designed an optimal circuit design link budget and a high-voltage differential-topology RF power amplifier.

## 2. Circuit Design Specification

Table 1 summarizes the maximum output power level according to the UE power class, signal BW, and modulation order. A tolerance of ±2 dB is generally required as high output power can interfere with other channels, and insufficient maximum power can reduce all cell coverage. To meet the power class 2 (PC2) output power and support 100 MHz ET operation, this sub-section describes two circuit specifications: (1) PA saturation power generation, and (2) maximum PA supply voltage and power capacity.

The RF power amplifier should be able to efficiently generate the target saturation power according to Equation (1). To generate the desired saturation power of Psat.PA, the PA power cell ($P_{sat.PAcell \mid dBm}$) has to generate higher saturation power to compensate for the output transformer loss ($\eta_{XFMR \mid dB}$) and intrinsic loss ($\eta_{int \mid dB}$).

**Table 1.** Summary of TX power requirements.

| Condition | | CP-OFDM (PAPR = 9 dB) | | | DFT-s-OFDM (PAPR = 6 dB) | | |
|---|---|---|---|---|---|---|---|
| Mod. | RB | MPR (dB) | PC3 (dBm) | PC2 (dBm) | MPR (dB) | PC3 (dBm) | PC2 (dBm) |
| QPSK | Inner | 1.5 | 21.5 | 24.5 | 0 | 23 | 26 |
| | Outer | 3 | 20 | 23 | 1 | 22 | 25 |
| 16QAM | Inner | 2 | 21 | 24 | 1 | 22 | 25 |
| | Outer | 3 | 20 | 23 | 2 | 21 | 24 |
| 64QAM | Outer | 3.5 | 19.5 | 22.5 | 2.5 | 20.5 | 23.5 |
| 256QAM | Outer | 6.5 | 16.5 | 19.5 | 4.5 | 18.5 | 21.5 |

The maximum current ($I_{max}$), which determines the size of the power cell, and the optimum impedance ($R_{opt}$) are obtained from the following equations [10].

$$P_{sat.PA|dBm} = P_{TX.ant|dBm} + PAPR + IL_{FEM|dB} \tag{1}$$

$$P_{sat.PAcell|dBm} = P_{sat.PA|dBm} + \eta_{int|dB} + \eta_{XFMR|dB} \tag{2}$$

$$\eta_{XFMR.PA} = \frac{1}{1 + \frac{r}{Q_{ind}^2}}, \quad r = \frac{50}{R_{opt}} \tag{3}$$

$$i_d(\theta) = \frac{I_{max}}{(1 - \cos(\alpha/2))}[\cos\theta - \cos(\alpha/2)] \tag{4}$$

$$I_{dc} = \frac{1}{2\pi} \int_{-\alpha/2}^{\alpha/2} \frac{I_{max}}{(1 - \cos(\alpha/2))}[\cos\theta - \cos(\alpha/2)]d\theta \tag{5}$$

$$I_n = \frac{1}{\pi} \int_{-\alpha/2}^{\alpha/2} \frac{I_{max}}{(1 - \cos(\alpha/2))}[\cos\theta - \cos(\alpha/2)]\cos n\theta d\theta \tag{6}$$

$$I_1(A) = \frac{2 \cdot P_{sat.PAcell}}{V_{DD} - V_{knee}}, \quad R_{opt} = \frac{V_{DD} - V_{knee}}{I_1} \tag{7}$$

$$I_{max} = \frac{2 \cdot \pi \cdot I_1 \cdot \left[1 - \cos\left(\frac{\alpha}{2}\right)\right]}{\alpha - \sin(\alpha)}, \quad I_{dc} = \frac{I_{max}}{2\pi} \cdot \frac{2\sin\left(\frac{\alpha}{2}\right) - \alpha\cos\left(\frac{\alpha}{2}\right)}{1 - \cos\left(\frac{\alpha}{2}\right)} \tag{8}$$

$$I_q = I_{max} \cdot \frac{\cos\left(\frac{\alpha}{2}\right)}{\cos\left(\frac{\alpha}{2}\right) - 1} \tag{9}$$

where $\alpha$ is the conduction angle, $i_d(\theta)$ is the RF current waveform, $I_{dc}$ is the mean current, $I_n$ is the magnitude of the nth harmonic, and $I_1$ is the fundamental component of the conducted current at the power cell. In general, a quiescent current does not change according to the input signal level, but the mean (dc) current changes according to the input voltage level as shown in Figure 1a. As shown in Figure 1b, the conduction angle, $\alpha$, of the ideal class-B PA is maintained at $\pi$ in all output power regions, but in a real PA with a fixed quiescent bias current, $I_q$, it increases the conduction angle ($\alpha_{PBO}$) at the power back-off. After fixing the quiescent bias current, a dc current at the linear power back-off ($P_{PBO}$) as high as the PAPR from the saturation power is obtained from the following equations.

$$I_q = \frac{4 \cdot \pi \cdot P_{sat.PAcell}}{V_{DD} - V_{knee}} \frac{1 - \cos\left(\frac{\alpha_{PBO}}{2}\right)}{\alpha_{PBO} - \sin(\alpha_{PBO})} \tag{10}$$

$$I_{dc.PBO} = \frac{2 \cdot (P_{sat.PAcell} - P_{PBO})}{V_{DD} - V_{knee}} \cdot \frac{2\sin\left(\frac{\alpha_{PBO}}{2}\right) - \alpha_{PBO}\cos\left(\frac{\alpha_{PBO}}{2}\right)}{\alpha_{PBO} - \sin(\alpha_{PBO})} \tag{11}$$

$$I_{dc.PBO} = I_q, \quad when \ \alpha_{PBO} = 2\pi \tag{12}$$

where $I_{dc.Psat}$ of the instantaneous peak current and $I_{dc.Plin}$ of the average current of PA load can be obtained from (11). As shown in Figure 1c, there are two supply power requirements

of average output power ($P_{SM.avg}$) and instantaneous peak output power ($P_{SM.peak}$), which have different values depending on the signal's PAPR. In the APT mode, the inductor average current ($I_{SA.avg}$) of the DC-DC converter is equal to the average PA load current ($I_{dc.Plin}$), and the AC current of the PA load current with a high-frequency component is provided by the output capacitor and the PA supply decoupling capacitor. Therefore, in order to minimize the AC ripple voltage at the PA supply node, capacitors with different self-resonant frequencies (SRFs) need to be connected in parallel to exhibit a low supply impedance in a wide frequency range. In the ET mode, the switching amplifier supplies the average output current ($I_{SA.avg}$), and the push–pull class-AB output buffer of the linear amplifier additionally supplies the AC current to provide instantaneous peak current ($I_{LA}$) and voltage ($V_{SMout}$) to the PA.

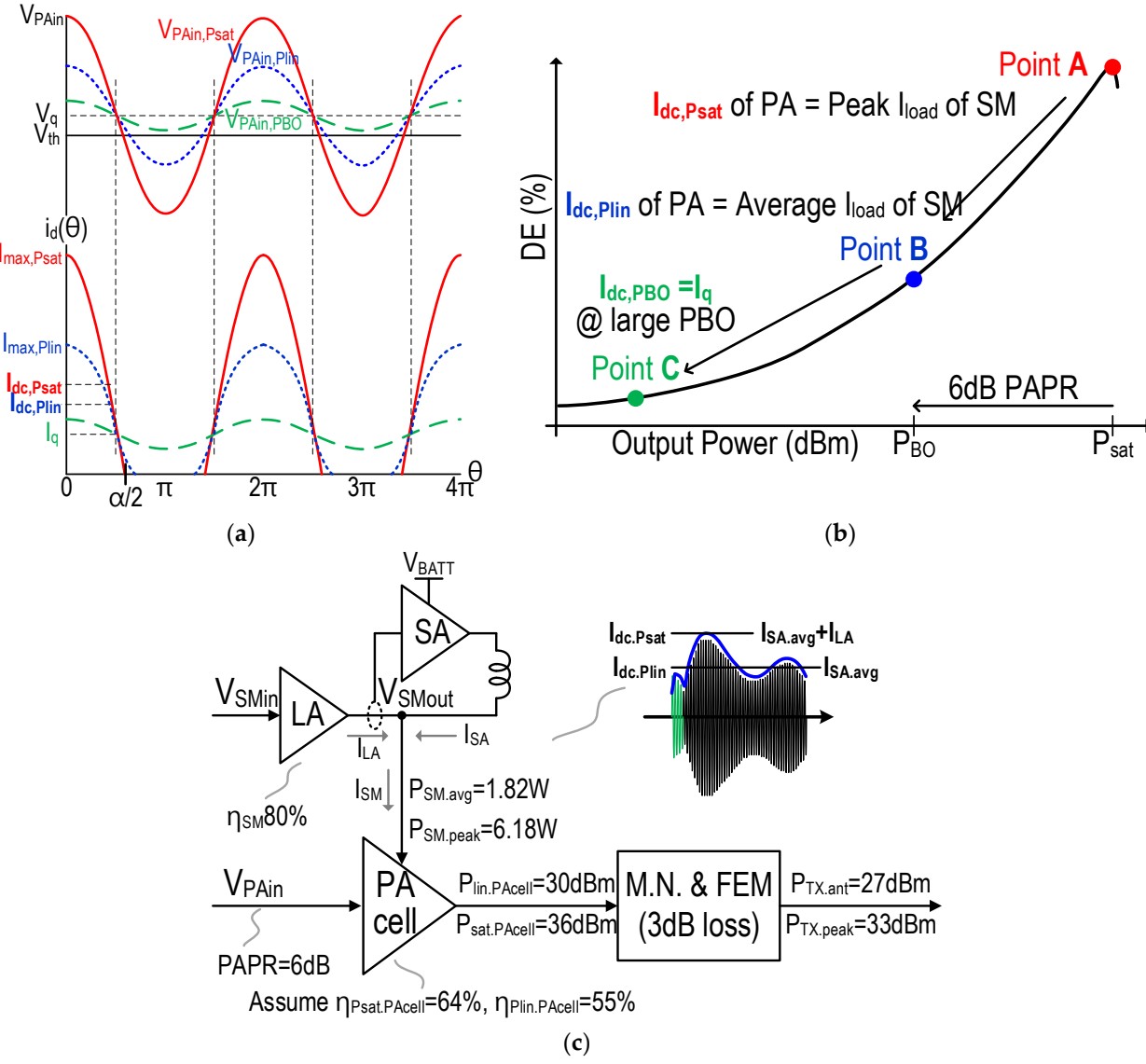

**Figure 1.** Analysis of SM-IC output current and power capability from RF-PA supply dc current analysis. (**a**) Current waveforms of the power cell according to input voltage level. (**b**) Drain efficiency curve versus PA output power. (**c**) Maximum output power capability of SM-IC.

Figure 2 shows an example of $R_{opt}$ and $I_{max}$ according to the PA topology and the required supply voltage to generate a target 36 dBm saturation power. In general, as the impedance transform ratio from a 50 $\Omega$ load to the desired optimum impedance ($R_{opt}$) increases, the loss of the output matching network also increases. Using the differen-

tial topology and 5 V boosted output supply voltage can reduce both the $I_{max}$ and the impedance transform ratio for generating 36 dBm of output power. A smaller power cell size can exhibit a smaller PA supply parasitic capacitance, so it is easy to extend the 3 dB BW of the SM required for 100 MHz ET operation.

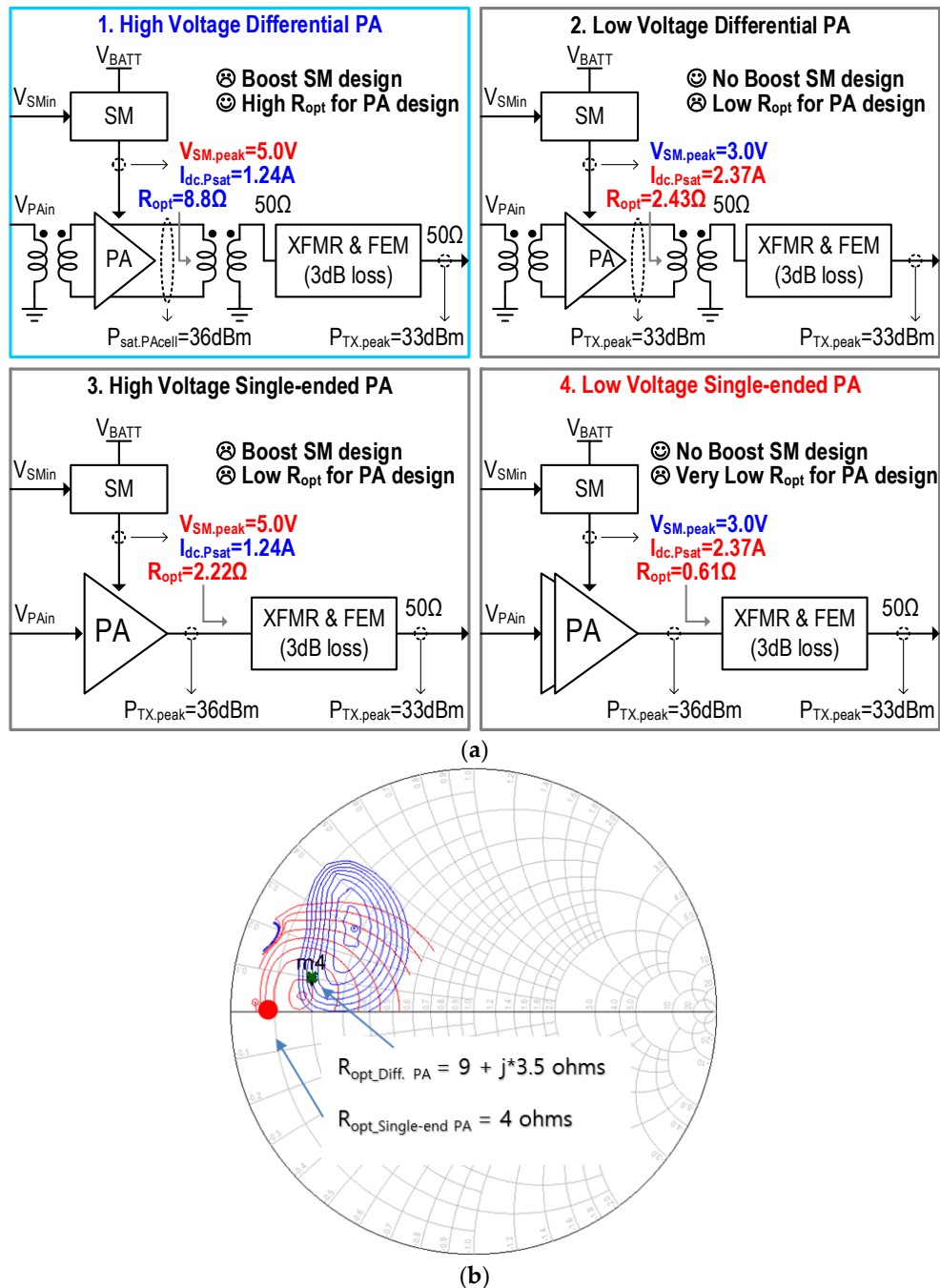

(a)

(b)

**Figure 2.** (**a**) $I_{max}$ and $R_{opt}$ comparison according to PA topologies and required supply voltage. A conduction angle of $1.05\pi$ at a $P_{sat.PAcell}$ of 36 dBm and a $V_{knee}$ of 0.8 V are assumed. (**b**) Load–pull contour simulation results.

## 3. Design of n77 HPUE Power Amplifier Module

The I/Q signal up-converted to the RF carrier frequency is amplified by the power amplifier, and unwanted spurious components are filtered by the RF bandpass filter and

finally sent to the antenna. The RF front-end (RFFE) should be designed to satisfy the TX power requirements summarized in Table 1.

Figure 3 shows the designed n77 band power amplifier module (PAM). The designed PAM consists of a high-voltage differential-topology [17] 2 μm GaAs HBT power amplifier, a CMOS controller supporting MIPI 2.1 RFFE, a silicon-on-insulator (SOI) switch, an integrated passive device (IPD) bandpass filter, a low-noise amplifier, and a bi-directional coupler. In order to obtain a saturation power of 35.7 dBm with a 5 V supply voltage at a 3.75 GHz center frequency, the optimum impedance $Z_{opt}$ is designed as 9 + j3.5 Ω, and the current density of the power cell is 0.3 mA/μm². The input and inter-stage matching network are implemented with an on-chip transformer (XFMR), and the output XFMR is designed on a PCB for low substrate loss and transforms the 50 Ω load into the desired optimum load, $R_{opt}$. The quality factor, Q, of the output XFMR's primary and secondary inductors are typically 45 and 35, respectively. To improve the Q of the primary side through which DC current and AC current flow greatly, the primary inductor was designed as a top metal, and the secondary inductor was designed as a vertical type using the 2nd/4th layer to improve the coupling factor, k. RF overvoltage protection (OVP) and DC OVP circuits are applied to prevent device breakdown by an antenna mismatch and SM output overvoltage, respectively. Basically, the SM output in the ET mode is regulated as DAC output to prevent overvoltage, but the ET OVP circuit [8] is also applied to prevent the overvoltage over 5.5 V at the SM output even in abnormal conditions.

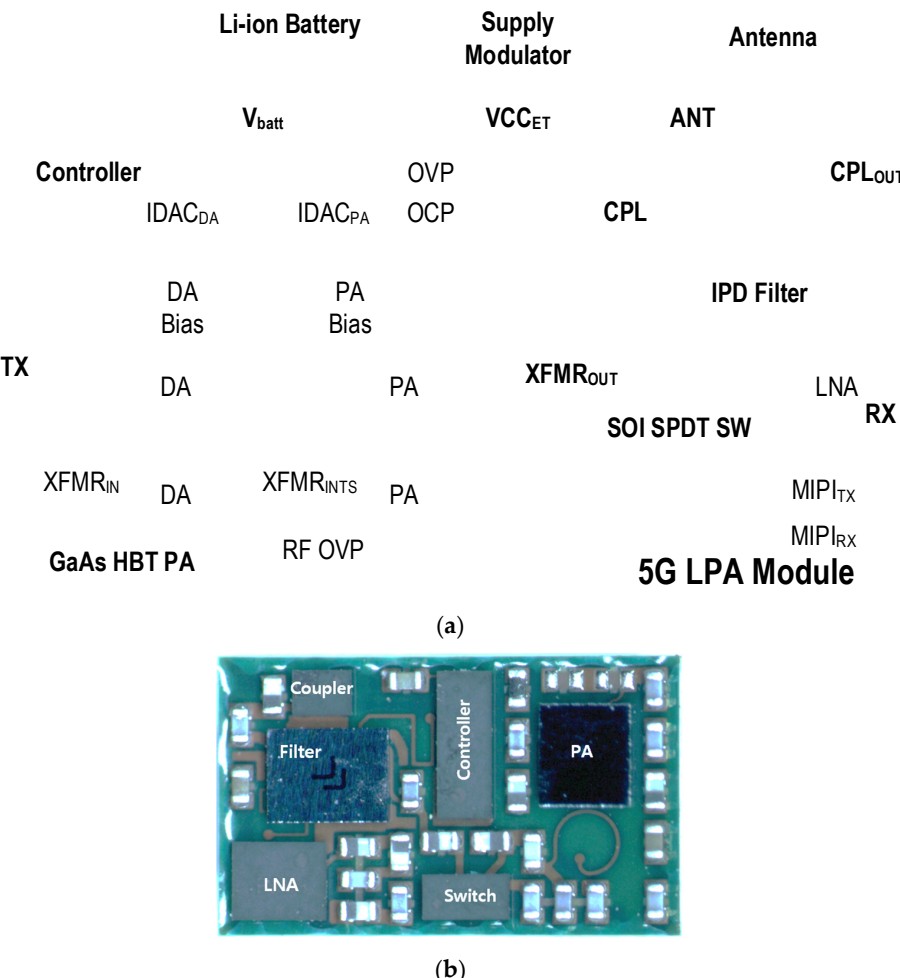

**Figure 3.** A 5G n77 low-noise amplifier and power amplifier (LPA) module diagram consisting of RF-PA, CMOS controller, SOI switch, IPD bandpass filter, coupler, and low-noise amplifier. (**a**) Block diagram. (**b**) Module photograph.

Figure 4 shows the designed bandpass filter by using an integrated passive device (IPD) process. The filter size is $1.6 \times 0.9$ mm$^2$.

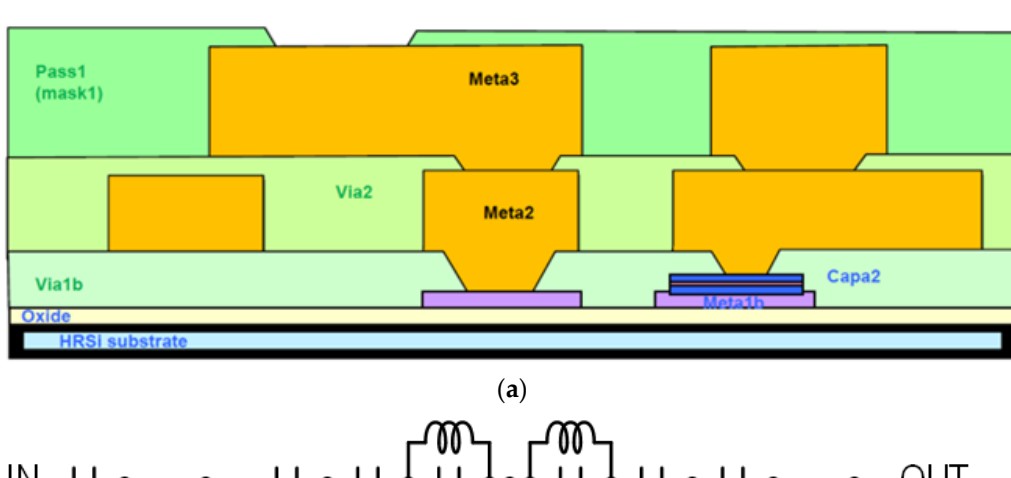

(**a**)

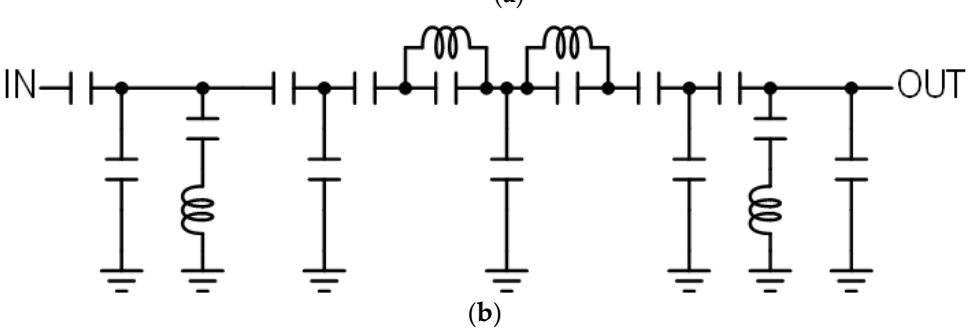

(**b**)

**Figure 4.** (**a**) IPD bandpass filter process. (**b**) Designed n77 bandpass filter.

The RF switch is designed with an asymmetric single-pole double-throw (SPDT) structure that optimizes the TX path loss, and 0.3 dB and 0.5 dB insertion losses are obtained for the TX and RX paths, respectively. It has 2nd/3rd harmonics of less than $-60$ dBm at a 32.5 dBm output power and achieves an isolation of 35/30 dB at 4.2 GHz for the antenna-to-TX and antenna-to-RX, respectively. The RF switch achieves more than 39 dBm of $P_{0.1dB}$. Figure 5a shows the SPDT switch and Figure 5b shows the die micrograph of the designed GaAs HBT PA.

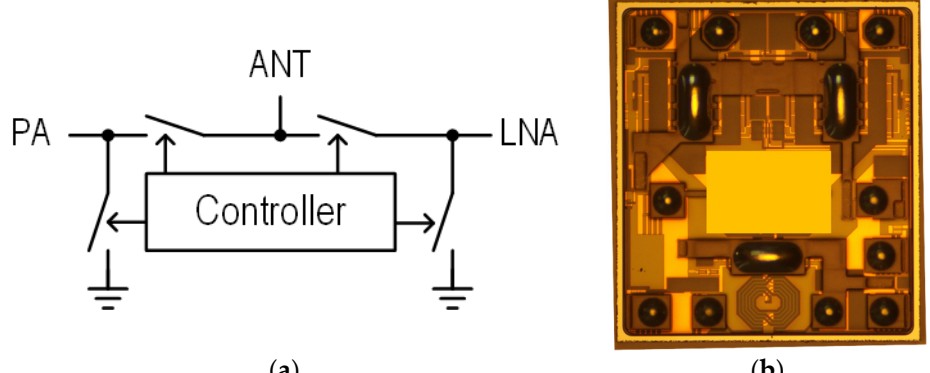

(**a**)          (**b**)

**Figure 5.** (**a**) Asymmetric single-pole double-throw (SPDT) switch. (**b**) Die micrograph of designed GaAs HBT PA.

## 4. Experimental Results

Figure 6a shows the frequency response of the designed bandpass filter. It has 2 dB of typical insertion loss and $-55$ dB of attenuation for an EN-DC coexistence scenario with LTE band 1 and band 3. Figure 6b shows the overall frequency response of the gain and saturation power of the designed PAM. The designed PAM achieved a typical small signal

gain and a saturation power of 30.42 dB and 32.7 dBm at 3.7 GHz of center frequency, respectively. In addition, the gain and power variation of 1.33 dB and 1.5 dB were achieved, respectively, in the n77 band excluding the upper edge frequency of 4.2 GHz. The total PA module size was $3 \times 5$ mm$^2$.

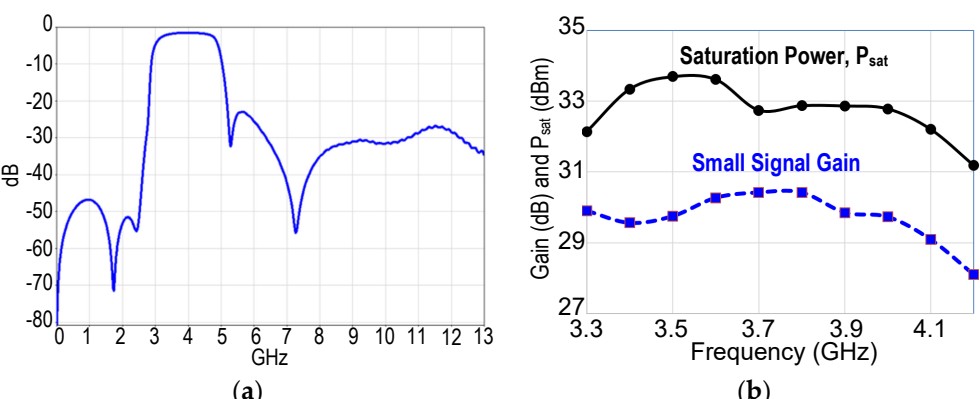

**Figure 6.** (**a**) Bandpass filter frequency response, S21. (**b**) Frequency response of the gain and saturation power of the designed PAM.

DFT-spread-OFDM QPSK and CP-OFDM 256QAM-modulated signals are used in the ET TX system validation. DFT-s-OFDM and CP-OFDM signals, which have passed through the CFR filter in the digital front end, have PAPRs of 6 dB and 8 dB, respectively. The 5G n77 ET-PA achieves a $-41/-38.4$ dBc ACLR at a 26 dBm output power with a DFT-s-OFDM QPSK 100 MHz NR signal. For the DFT-s-OFDM QPSK with the most stringent output power requirements, ACLR and EVM with less than $-37$ dBc and 1.54% were achieved at an output power of up to 27 dBm, respectively. The 26 dBm of the CP-OFDM QPSK signal consumes a similar dc power to the 27 dBm DFT-s-OFDM QPSK signal to maintain a similar linearity of $-38$ dBc because the PAPR is 2 dB higher than the DFT-s-OFDM QPSK signal. For the CP-OFDM 256QAM with the most stringent EVM requirements, it achieves an EVM of 1.22% at 23 dBm and an output power margin 3.5 dB higher than the standard specification of 19.5 dBm. In addition, it saves a dc power consumption of 640 mW compared to APT PA. Figure 7 shows the measured EVM and ACLR of the ET-PA according to the operating frequency. Table 2 summarizes the ET-PA performance according to each modulation waveform. The overall performance comparison with prior works is shown in Table 3.

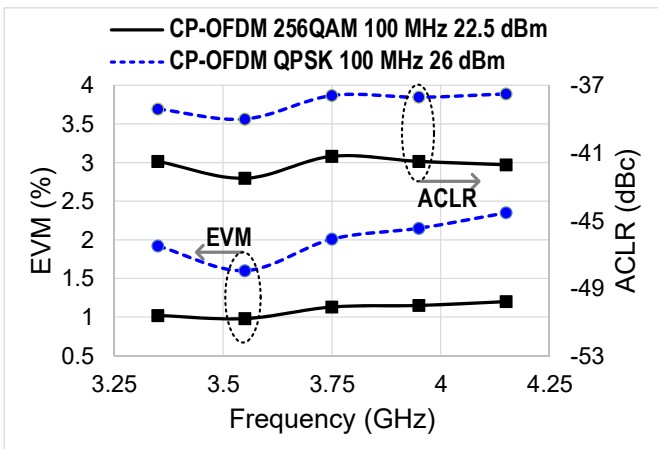

**Figure 7.** Measured EVM and ACLR according to the operating frequency in the n77 band.

**Table 2.** ET-PA performance summary.

| Modulation | Supply Mode | Pout (dBm) | ACLR (dBc) | $P_{dc}$ * (W) | EVM (%) |
|---|---|---|---|---|---|
| DFT_QPSK | APT | 27 | −37.1 | 4.10 | 1.40 |
| 100 MHz | ET | 27 | −37.0 | 3.15 | 1.54 |
| CP_QPSK | APT | 26 | −38.2 | 4.13 | 1.27 |
| 100 MHz | ET | 26 | −38.4 | 3.17 | 1.29 |
| DFT_256QAM | APT | 24.5 | −38.2 | 2.54 | 1.6 |
| 100 MHz | ET | 24.5 | −38.3 | 2.09 | 1.34 |
| CP_256QAM | APT | 23 | −42.0 | 2.54 | 1.50 |
| 100 MHz | ET | 23 | −41.5 | 1.9 | 1.22 |

* dc power consumption is measured at 100% duty operation.

**Table 3.** Performance comparison.

| Ref. | Protocol | BW (MHz) | Center Freq. (GHz) | Pout (dBm) | ACLR (dBc) |
|---|---|---|---|---|---|
| This Work | 5G NR | 100 | 3.75 | 27.4 | −37.1 |
| [19] ISSCC16 | 4G LTE | 40 | 1.95 | 24 | −41.2 |
| [20] ISSCC19 | 5G NR | 100 | 3.5 | 23 | −38 |
| [21] ISSCC18 | 4G LTE | 80 | 2.59 | 26 | −38.2 |

## 5. Conclusions

This paper presents a highly efficient 5G NR n77 power amplifier module. The new 5G UE RF transmission requirements are summarized and the optimized circuit link budget between the RF power amplifier and envelope tracker output is analyzed. Based on the link budget design, the n77 PAM is optimally designed and implemented. The ET PA achieves the target output power and linearity while saving battery power consumption.

**Funding:** This paper was supported by a Korea Institute for Advancement of Technology (KIAT) grant funded by the Korea Government (Ministry of Education) (P0025688, Semiconductor-Specialized University).

**Data Availability Statement:** Data are contained within the article.

**Conflicts of Interest:** The author declares no conflicts of interest.

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
