# Peer review of "n77 Radio Frequency Power Amplifier Module for 5G New-Radio High-Power User Equipment Mobile Handset Applications"

_electronics, doi:10.3390/electronics13050908_

Round 1

Reviewer 1 Report

Comments and Suggestions for Authors

There are significant issues with this paper, particularly in the lack of detail. It is not clear in the abstract what the novelty is. Largely, it needs more details. For example what Fig. 2 is not clear. There are no schematics for either the PA or the supply modulator which is described as being a commercial one, but of which no details are given. Similarly for the BP filter, for which results are presented, but no details given. If there is novelty here, it is not clear at all.

Author Response

I appreciate your contribution to review this paper.

I added more details in Fig. 2, Fig. 4, Fig. 5, and Table III.

Thanks

Best regards

Jiseon

Reviewer 2 Report

Comments and Suggestions for Authors

This paper presents a power amplifier module for 5G NR n77 band handset applications. Please find my comments as follows:

1. More information expressing the reseach background is needed for the introduction. For instance, published configurations of the module can be included.

2. Specifications of the individual building blocks such as SPDT switch must be presented.

3. Simulation results are required to be included with the measured ones.

4. Comparison with commercial products or published works are required.

Author Response

I appreciate your contribution to review this paper.

  1. I added detailed circuit implementations in new Figure 4 and Figure 5.
  2. Comparison table III is updated. 
  3. RF PA load pull simulation is added to verify the match with hand calculation result. 

Thanks

Best regards

Jiseon

Reviewer 3 Report

Comments and Suggestions for Authors

In this work, the authors proposed a highly efficient 5G New Radio (NR) RF power amplifier module (PAM) for mobile handset applications. The module integrates a high-voltage differential topology, a CMOS controller, an SOI switch, an integrated passive device (IPD) bandpass filter, a low noise amplifier (LNA), and a bi-directional coupler. It exhibits significant improvements in terms of output power, linearity, and power consumption, specifically tailored for the n77 band of 5G NR. This work is of interest, however there are some concerns of this reviewer to be addressed. Please find below my comments:

- How does the high-voltage differential topology in the PAM contribute to its overall efficiency, particularly when compared to conventional PAM designs?

- Can you provide more details on the design and operation of the CMOS controller, especially regarding its interaction with other components of the PAM?

- The paper mentions the integration of a silicon on insulator (SOI) switch. Can you elaborate on the choice of SOI technology and its benefits in this application?

- The IPD bandpass filter is a critical component. Could you discuss the design considerations and how this filter maintains performance across the specified frequency range?

- For the low noise amplifier (LNA), what specific design strategies were employed to minimize noise while ensuring amplification is in line with the rest of the module's components?

- The bi-directional coupler's design and its role within the PAM are not extensively covered. Could you provide more insights into its design and how it enhances the module's performance?

- In terms of the saturation output power and ACLR performance, how does the PAM compare with existing solutions in the market, and what makes it stand out?

- Power consumption is a critical aspect, especially for mobile handset applications. How does the PAM's power consumption compare to other contemporary solutions, and what design elements contribute to its efficiency?

- Can you discuss the testing methodology used to validate the PAM's performance, particularly how the commercial envelope tracker IC (ET-IC) was integrated during testing?

- Lastly, are there any scalability considerations or future improvements planned for the PAM to adapt to evolving 5G NR standards or to cater to different frequency bands?

Comments on the Quality of English Language

Moderate English language edits are required. 

Author Response

I appreciate your contribution to review this paper.

Please find my feedback below

- How does the high-voltage differential topology in the PAM contribute to its overall efficiency, particularly when compared to conventional PAM designs?

  : High voltage differential topology can provide higher optimum resistance than the conventional PA topology. It can reduce the impedance transform ratio, and overall matching network loss can be minimized. The better matching network loss improves the overall power efficiency of the RF PA. Furthermore, high VCC can provide a larger voltage swing level (VCC-Vknee) when we assume same Vknee voltage. I can help to get higher efficiency in the RF PA design.

- Can you provide more details on the design and operation of the CMOS controller, especially regarding its interaction with other components of the PAM?

 : We used the MIPI interface to control the PAM. MIPI 3.0 protocol is typically used serial to parallel digital interface for the RFFE control.  

- The paper mentions the integration of a silicon on insulator (SOI) switch. Can you elaborate on the choice of SOI technology and its benefits in this application?

: Actually, the SOI switch has lower substrate loss than the buck CMOS process. So, we selected the SOI technology for this project. 

- The IPD bandpass filter is a critical component. Could you discuss the design considerations and how this filter maintains performance across the specified frequency range?

 : Low insertion loss is very important. Additionally, low-side filter isolation performance is important to avoid some problems by co-existence with LTE band1 and band3.  And, conventional SAW and BAW filters are too expensive solutions. So, we need to get a cheaper solution like the IPD device. 

- For the low noise amplifier (LNA), what specific design strategies were employed to minimize noise while ensuring amplification is in line with the rest of the module's components?

: There are no specific circuit techniques for LNA design.  

- The bi-directional coupler's design and its role within the PAM are not extensively covered. Could you provide more insights into its design and how it enhances the module's performance?

  : The bi-directional coupler design itself doesn't improve the module's performance. It senses the RF PA output signal, and it feedback to the baseband modem for the digital pre-distortion. By the DPD operation, we can improve the RF power amplifier's linearity.  

- In terms of the saturation output power and ACLR performance, how does the PAM compare with existing solutions in the market, and what makes it stand out?

  : It's hard to compare with existing solutions in the market. 

- Power consumption is a critical aspect, especially for mobile handset applications. How does the PAM's power consumption compare to other contemporary solutions, and what design elements contribute to its efficiency?

 : Due to the integrated filter, we cannot compare the power amplifier's efficiency (like the PAE). So, we compare the total dc power consumption of the PAM or PAMid. 

- Can you discuss the testing methodology used to validate the PAM's performance, particularly how the commercial envelope tracker IC (ET-IC) was integrated during testing?

  : Actually, I have the ET-IC which is an available commercial product. 

- Lastly, are there any scalability considerations or future improvements planned for the PAM to adapt to evolving 5G NR standards or to cater to different frequency bands?

: Several technologies will be available. Digital envelope tracking, Doherty PA with supply modulation, Symbol power tracking are all good candidates for next PAM technology.

Thanks

Best regards

Jiseon

Round 2

Reviewer 1 Report

Comments and Suggestions for Authors

This paper is improved, but a more detailed description and schematic of the PA is required. Similarly, details of the supply modulator must be included.

Author Response

Thank you for your effort.

Actually, this paper used a conventional hybrid linear-assisted supply modulator for an ET-link test. 

Detailed circuit topology of the supply modulator is not in the scope of this paper.

Thanks

Best regards

Jiseon

Reviewer 2 Report

Comments and Suggestions for Authors

The reviewer thanks for the efforts that the author has spent on addressing my concern. The issues are clearly addressed and the paper can be accept in its current form. 

Author Response

Thank you for your effort.

Best regards

Jiseon

Reviewer 3 Report

Comments and Suggestions for Authors

Thank you for addressing the concerns of this reviewer. I have no more comments. 

Author Response

(The authors gave the same response as above.)

Round 3

Reviewer 1 Report

Comments and Suggestions for Authors

From the authors insufficient response to my question, I can only assume it is the Qualcom ET modulator that they are using.